# Numerical Modelling of the Mechanical Behaviour of Biaxial Weft-Knitted Fabrics on Different Length Scales

**DOI:** 10.3390/ma12223693

**Published:** 2019-11-08

**Authors:** Minh Quang Pham, Oliver Döbrich, Wolfgang Trümper, Thomas Gereke, Chokri Cherif

**Affiliations:** Technische Universität Dresden, Faculty of Mechanical Science and Engineering, Institute of Textile Machinery and High Performance Material Technology (ITM), 01062 Dresden, Germany; oliver.doebrich@tu-dresden.de (O.D.); wolfgang.truemper@tu-dresden.de (W.T.); thomas.gereke@tu-dresden.de (T.G.); chokri.cherif@tu-dresden.de (C.C.)

**Keywords:** composite, draping, finite element method, forming, macro-scale model, meso-scale model, weft-knitted fabric

## Abstract

Weft-knitted fabrics offer an excellent formability into complex shapes for composite application. In biaxial weft-knitted fabric, additional yarns are inserted in the warp (wale-wise) and weft (course-wise) directions as a reinforcement. Due to these straight yarns, the mechanical properties of such fabrics are better than those of unreinforced weft-knitted fabrics. The forming process of flat fabrics into 3D preforms is challenging and requires numerical simulation. In this paper, the mechanical behavior of biaxial weft-knitted fabrics is simulated by means of macro- and meso-scale finite element method (FEM) models. The macro-scale modelling approach is based on a shell element formulation and offers reasonable computational costs but has some limitations by the description of fabric mechanical characteristics and forming behavior. The meso-scale modelling approach based on beam elements can describe the fabric’s mechanical and forming characteristics better at a higher computational cost. The FEM models were validated by comparing the results of various simulations with the equivalent experiments. With the help of the parametric models, the forming of biaxial weft-knitted fabrics into complex shapes can be simulated. These models help to predict material and process parameters for optimized forming conditions without the necessity of costly experimental trials.

## 1. Introduction

Composites made of continuous fiber reinforced polymers (FRP) have been increasingly researched and used to reduce the energy consumption of various means of transport. Components made from FRP have significant lower weight with the same or enhanced mechanical properties in comparison to metallic components. FRPs are preferentially manufactured based on a thermoset matrix (more than 75% of all the composites) due to the ease of manufacturing, higher thermal stability, excellent fatigue strength, and good fiber to matrix adhesion [1]. FRP based on a thermoplastic matrix are attracting growing interest because of unlimited storage, semi-products delivered ready for use, thermoformability, fast consolidation, and environmental friendliness. Thermoplastic composites can be made from fully impregnated organic sheets or partially impregnated composite fabrics using polymer powders, solvent impregnation, dipping or coating with molten matrix, coating with films or nonwoven fabrics, and insertion of thermoplastic yarns or hybrid yarns made of reinforcement fibers and thermoplastic fibers [2].

Hybrid yarns have several advantages. The thermoplastic polymer matrix, which is blended with the reinforcing fiber into the hybrid yarn by means of an online melt spinning technology [3] or commingling technology [4], is melted in a thermoforming process and solidified. Since the reinforcing and the thermoplastic components are combined within one hybrid yarn, the flowing distance of the thermoplastic polymer matrix to the reinforcing fibers is significantly reduced. Hence, the cycle time is reduced, and the impregnation is improved.

Biaxial fabrics are reinforced in the length and the width of the fabric. The presence of straight or undulated yarns significantly affects their mechanical behavior. Weft-knitted fabrics offer an excellent formability into complex shapes. In biaxial weft-knitted fabric, yarns are inserted in the warp (wale-wise) and weft (course-wise) directions as a reinforcement [5,6,7,8]. Due to these straight yarns, the mechanical properties of such fabrics are better than those of unreinforced weft-knitted fabrics with the forming capacity still being high. The mechanical properties and the forming properties can be adjusted by the choice of the fiber material, the yarn cross sections, yarn distances, amount of weft and warp layers used. However, numerical models are missing in order to predict their mechanical behavior, which is important for composite applications.

One of the challenges of the FRP production process lies in the forming process of 2D fabrics to 3D preforms. During this forming process, many unexpected defects may occur. One of the most encountered defects is the formation of wrinkles within the load bearing area of the preform, which would reduce the mechanical properties of the composite component [9]. Other defects such as breakage, sliding, in-plane and out-out-plane buckling of yarns, and gap formation also have a great impact on the quality of the final component. At high blank holder forces, the friction between textile and forming tools and interlaminar friction between fabric layers induce local in-plane tensile forces into the textile [10]. The forming tool geometry and forming process parameters can be adapted in order to manipulate the magnitude, direction, and distribution of tensile force in order to suppress the forming defects or to reach a load path optimized fiber orientation [11].

Process design by “trial and error” method costs time and material. For example, changing the geometry of the forming tools would require a long lead time and iterative forming experiments to optimize the machine and process parameters, which would waste an enormous amount of material. A simulation approach would reduce the developing time and cost. Therefore, many models for various textile structures such as woven fabrics [9,12,13,14,15,16,17,18,19,20,21,22,23,24,25,26,27,28,29], 3D woven fabrics [30,31,32,33,34,35,36], non-crimp fabrics [37], weft-knitted fabrics [17,38,39,40,41,42,43,44,45,46,47], and multiaxial warp-knitted fabrics [48] were developed. The models could be classified by the mathematical approach: Analytical [44,49,50] or numerical models, in which they are divided into smaller groups by the length scale of the modelled components into three categories: Micro-scale models [20,21,35,36,39], meso-scale models [16,17,18,19,22,31,32,33,34,37,40,41,42,44,45,46,47,49], and macro-scale models [10,13,14,15,16,23,24,25,26,27,28,29,30,31,39].

Micro-scale models are closest to reality. The smallest unit to be modelled is the filament. For example, a multifilament yarn is represented by a set of many filaments that are modelled with beam element (Figure 1). Although the quantity of beam element chains is normally far smaller than the quantity of filament in the real yarn, a good representation can still be achieved [21]. However, extremely high computational costs have limited the application of micro-scale models. They are mostly used to predict the mechanical behavior of a new textile structure with known material characteristics of the yarn, where only a unit cell of fabrics structure must be modelled.

In meso-scale models, the smallest modelled unit is the yarn, which can be represented by beam, shell, or solid elements. This method still belongs to the discrete approach and the interaction between yarns in the whole textile structure is considered. By this simplified modelling method, meso-scale models have significant lower computational cost compared to micro-scale models, and this allows large scale simulations for the textile production process and for the forming process (Figure 2). Most of the important forming mechanisms can be described. The results of such forming process simulations help to optimize the process virtually as well as prepare information for further structure analysis of the composite part. At the yarn level, phenomena such as yarn sliding, gap formation, and yarn breakage are predictable.

Macro-scale models are based on continuum mechanics and are applicable for a wide range of structures. As the textile structure is assumed to behave as continuum, the geometrical configuration of the yarns in the structure, as well as the interaction between yarns, is not directly modelled. The mechanical behavior of the whole textile structure is modelled by suitable material laws for the continuum. Mechanical properties, such as tensile, shear, and bending behavior, are determined by physical or virtual tests. Due to the homogenization of the fabric structure, macro-scale models have the least computational cost. However, defects such as wrinkle formation, fiber orientation, and textile failure can be predicted on a macroscopic level (Figure 3).

A few models based on the finite element method (FEM) for weft-knitted fabrics and biaxial reinforced weft-knitted fabrics were presented with different approaches. The commercial software WiseTex allows to model and visualize the geometries of many weft-knitted fabrics as 3D objects on the meso-scale [41]. These models can be further exported to third party software to be used in FEM packages to create FEM models of fabrics, predictive models of composites mechanics, or predictive models of textile permeability. De Araújo [43,49,50,51,52] introduced three models for the single jersey structure, one analytical model and two FEM models. The first model of de Araújo is a 3D analytical model based on the elastic theory, which can be used to calculate the mechanical behavior of the fabrics in the course-wise and wale-wise directions [50]. By the second attempt, a simple meso-scale model of 2D hexagonal FEA model with non-linear truss elements are used as a substitute for the knitting loop structure [43]. The input parameter of the truss elements is taken from experimental results and the elongation deformation of a planar fabric to fit a 3D spherical mold can be calculated. The third model is a 3D FEM model with a mesh of tetrahedral elements [49]. The yarn has a solid representation and a mechanical simulation is applied to obtain a 3D shaped loop. A composite unit cell with 3D tetrahedral elements coupled with a matrix mesh helps to predict the mechanical behavior of the weft-knitted reinforced composite. Similar approaches using a meso-scale model on a repeated unit cell are also used for numerical analysis of the mechanical behavior of the weft-knitted structure [40,42,46] or weft-knitted reinforced composite structure [38,47,53,54]. The obtained properties can be transferred to macro-scale model for further analysis. Duhovic [39] has explicitly simulated the knitting process to obtain the forced-determined geometry of yarns and their residual stresses incurred during the knitting process. Further analysis of the deformation mechanisms of composite is based on the obtained geometry of yarns in the knitting process simulation. In general, multi-scale modelling methods are used progressively to analyze the mechanical behavior of textile as well as textile reinforced composite.

In contrast to previous papers on similar thermoplastic composites [55,56,57,58], the objective of this study is to analyze and understand the complexity of the forming behavior of biaxial reinforced weft-knitted fabrics made of commingled yarns from carbon and polyamide 6.6 (PA 6.6) fibers. Therefore, numerical approaches on the finite element method (FEM) are presented that model the mechanical behavior of knitted fabrics under tensile, shear, and bending loads as well as friction experiments. Meso- and macro-scale approaches are analyzed and compared. A continuum mechanics approach for macro-scale models and a new geometrical modelling method to integrate complex fiber entanglements for meso-scale model are applied. The models are validated through experimental tests. Based on the validated models, the forming process can be optimized virtually and then practically implemented.

## 2. Materials and Physical Testing

### 2.1. Yarn Manufacturing and Testing

Commingled hybrid yarns were used as the reinforcement in the weft and warp directions. Carbon fiber (CF) with 200 tex of manufacturer Teijin Carbon Europe GmbH (Wuppertal, Germany) and polyamide 6.6 (PA 6.6) with 94 tex of manufacturer W. Barnet GmbH & Co. KG (Aachen, Germany) were used. Hybrid yarns were produced on an air jet texturing machine according to [4]. Process parameters were as follows: Overfeed of CF 2%, overfeed of PA 6.6 3.5%, air pressure 3.5 bar. The 1200 tex reinforcing yarns were formed by folding four comingled hybrid CF/PA 6.6 yarns, which had a fineness of 300 tex each. Folded yarns made of glass fiber (GF) with 2 × 68 tex and PA 6.6 with 94 tex were used as the knitting yarns. 

The linear density of the reinforcing hybrid yarns was measured by the skein method according to the standard DIN EN ISO 2060 [59]. Tensile tests of comingled hybrid yarns CF/PA 6.6 and the folded yarns of GF/PA 6.6 were carried out according to the standard ISO 3341 [60]. While the folded yarns of GF/PA 6.6 showed linear elastic behavior (Figure 4a), the comingled hybrid yarns from CF/PA 6.6 showed a non-linear elastic behavior at small strains. This is followed by a linear elastic zone (Figure 4b). This effect of the hybrid yarn is caused by the undulation of the fibers of both components, which were opened and mixed during the texturing process inside the air jet unit. This was necessary to reach a homogenous mixing of the two components [4]. Yarn properties are summarized in Table 1.

### 2.2. Fabric Manufacturing

Biaxial reinforced weft-knitted fabrics were fabricated on a modified flat-bed knitting machine Aries 3D (Steiger Participations Sa., Vionnaz, Switzerland). The machine has additional guiding systems for reinforcing yarns [6]. By configuring the process parameters on the knitting machine, namely the knitting loop length, the knitting speed, and the fabric pull-out velocity, two variants of fabrics were produced. The reinforcing yarn density varied in the weft direction while it was constant in the warp direction. The configurations of the fabrics are shown in Table 2.

Images of the fabrics are presented in Figure 5, where 10 by 10 yarns in the fabrics are shown. Since the weft and warp densities in variant 1 equal each other, a quadratic fabric is resulting. In variant 2, the density of the reinforcing yarns in the weft direction was increased, such that the 10 × 10-unit cell has a rectangular shape. Microscopic images reveal more information about the structure of the fabric (Figure 6). The reinforcing yarns in variant 1 have elliptical cross-section while that is more circular for variant 2. An explanation for this is the larger interaction forces between the knitting yarn system and the reinforcing yarn systems, which is caused by the reduced loop length in variant 2 as measured according to the standard DIN EN 14970 [61]. This phenomenon also leads to a greater thickness and area mass density of variant 2 as measured according to standards ISO 5084 [62] and DIN EN 12127 [63], respectively.

### 2.3. Fabric Testing

Fabric tensile tests were performed on a tensile testing machine Zwick Z100 (Zwick GmbH & Co. KG, Ulm, Germany) with a nominal load up to 100 kN according to the standard DIN EN ISO 13934-1 [64]. The accuracy of the force sensor on the machine Zwick Z100 is 0.1%. The optical recording system for the length change (displacement) has an accuracy of 0.15%. Samples were 300 mm long and 50 mm wide and the distance between both clamps was 200 mm. Demgen Vulkollan flat clamps were used in combination with a pneumatic grip at a pressure of 40 bar. Testing speed was 20 mm/min.

Results of the tests are shown in Table 3 and Figure 10. The tensile strength of both variants is similar in the warp direction, whereas variant 2 has a significant higher tensile strength in the weft direction. This follows the higher reinforcing yarn density. The results of the tensile test show a non-linear characteristic of the textiles tensile behavior. The first reason of this non-linear behavior of the textile is the fact that the reinforcing yarns behave non-linear as mentioned above. Another reason is the undulation of the reinforcing yarn in the structure due to the interaction with the knitting yarns. 

The shear resistance of the fabrics was measured with a picture frame test with a fabric reference surface of 200 × 200 mm^2^ (Figure 7a). Total fabric size was 300 × 300 mm^2^. The picture frame was constructed according to the work of Orawattanasrikul [65], which uses needles along the frame side to fix the fabric. The picture frame was attached to a tensile test machine of Zwick with a nominal load of 2.5 kN. The shear force was measured with a sensor, which was attached next to the upper clamp of the tensile test machine. The sensor force of this machine Zwick has also an accuracy of 0.1%. The travel distance of the upper clamp was also recorded, and the shear angle was calculated according to [9]:(1)$$\gamma =\frac{\pi }{2}-2\cos^{-1}\left(\frac{\sqrt{2}L+d}{2L}\right)$$
where *γ* (rad) is the shear angle, *L* (mm) is the length of the frame side (here 200 mm), and *d* (mm) is the current travel distance of the upper clamp. Additionally, an Artec Eva 3D scanner of company Artec3D (Luxembourg-City, Luxembourg) was used to scan the out-of-plane wrinkling on the textiles at the position with shear angle of 40°. The Artec Eva 3D scanner has a 3D point accuracy up to 0.1 mm. Shear force is plotted against the calculated shear angle in Figure 11. The shear resistance behavior shows non-linear characteristics. With the same shear angle, variant 2 requires a significantly lower force than variant 1. This could be explained by the different interaction forces between knitting yarns and reinforcing yarns of two fabric variants as the consequence of different knitting loop length and reinforcing yarn density. The results of the 3D scanner show clear wrinkle forming, its shape, and the out-of-plane deflection of the fabrics (Figures 12 and 13).

The out-of-plane bending stiffness of fabrics was tested on a cantilever bending test machine ACPM 200 of the German producer Cetex according to the standard DIN 53362 [66]. The test configuration is shown in Figure 7b. A textile stripe with a width of 50 mm was automatically pushed over an edge until a laser system detected an interaction of the free end of the fabric with a plane that was oriented 41.5° to the horizontal. The overhang length was recorded and the bending stiffness per unit width was calculated as [67]
(2)$$B=g\frac{m}{l}{\left(\frac{l_{o}}{2}\right)}^{3}$$
with *B* (cNcm^2^) as the bending stiffness per unit width, *g* as the gravitational acceleration [cm/s^2^], *m* (g) as the mass of the textile stripe specimen, *l* (cm) as the length of the textile specimen, here 30 cm, and *l_o_* (cm) as the overhang length. The testing results showed that the fabric variant 2 (24 samples) has greater bending stiffness in both directions than the first variant (25 samples), see Table 3.

Another important parameter for the draping process simulation is the friction coefficient (static and kinetic) between the textile fabrics and the metallic forming tool, namely the stamp, the blank holders, and the female tool. To determine these friction coefficients, tests were carried out on a biaxial tensile testing machine (Zwick GmbH & Co. KG, Ulm, Germany) with a nominal load of 100 kN and two couples of perpendicular axes. The test principle is shown in Figure 7c. A textile stripe of dimension 200 × 50 mm^2^ was clamped between a pair of metallic pressers (made of high-alloy steel for through-hardening 1.2379) with a dimension of 150 × 50 mm^2^. The textile stripe was pressed by the pair of metallic pressers at one end with a force *F_N_* = 2000 N. The other end of the textile stripe was clamped, and the textile was pulled along its length direction. The pull force *F_R_* was recorded during the test and used to calculate the friction coefficient as follows
(3)$$\mu =\frac{F_{R}}{F_{N}}.$$

The static friction force is greater than the pull force F_R_ at the beginning of the test, so that the textile stripe has no motion. When the pull force overcomes the static friction, the textile stripe starts to slide between the clamps and the friction force may decrease, defining the kinetic friction. In this case, the static and kinetic friction coefficients were found to be almost equal (Figure 16), such that the model input parameters for both are the same. A friction coefficient of 0.15 (± 0.01) was determined for both fabric variants.

## 3. Modelling Biaxial Weft-Knitted Fabrics

The textile structure was modelled on two different levels: Macro-scale and meso-scale. The macro model is based on a model of Döbrich et al. [12], where the textile was considered as a continuum. The fabric was modelled in LS-DYNA using 4-node shell elements. The material behavior of the fabrics was captured using a laminate formulation with three integration layers. This enabled the decoupled description of the tensile and the bending stiffness and the mechanical behavior of the textile could be described correctly. An additional failure criterion was defined through the limitation of the shell element strain. Ultimate strain at the point of failure was taken from experimental data of the textile stripe test as shown in Table 3 and Figure 10. Any elements that exceed this strain limitation in any direction were automatically deleted. As the carbon fiber is brittle, this behavior is considered realistic for modelling material failure. The tensile and shear behavior of the textile was modelled by using the stress–strain curves of the tensile and shear tests as input parameters. Nonlinear curves were implemented as pair values of strain and stress, as measured. Shell elements of quadratic size (aspect ratio 1) were used in the models for both fabric variants despite the different reinforcing yarn density in the weft direction. The element size used for the simulations was determined in a sensitivity analysis.

Despite its simplicity, the macro-scale model can describe most of the forming mechanisms. However, for the forming of composite components with complex geometries, a higher degree of objectivity is required. The approach with continuum mechanics encounters difficulties to describe some important phenomena, such as the slippage between the fibers and the bending stiffness of the fibers [68]. As a matter of fact, textile structures are discrete in general and build up from smaller elements, namely the yarns, which are fixed together in the textile structure by their structural configuration. This architecture allows a relative movement between yarns, which can only be described by discrete FEM approaches.

A beam element approach initially presented in [45] was improved for modelling the biaxial reinforced weft-knitted fabrics on the meso-scale. To keep the computational cost affordable, only one beam element chain was used to represent each yarn. This discrete approach allows not only to describe the slippage between the yarns and the bending stiffness of the fibers, but also enables the failure at yarn level through the limitation of the beam element strain. The failure of the knitting yarns and the reinforcing yarns can be observed separately. The beams were modelled with simplified circular cross sections at the risk of decreasing accuracy of the model in the description of yarn to yarn surface contact. A linear elastic material model was used for the reinforcing and the knitting yarns [69]. The geometry of the knitting yarn was modelled by the mathematic equations of Choi and Lo [70]. The loop length of knitting yarn for each fabric variant can be accurately adapted. The simple linear elastic material model works well for the knitting yarn. It is assumed that the reinforcing yarns only have the waviness along their axial direction in a harmonic manner. Despite the linear elastic model, the non-linear tensile behavior of the reinforcing yarns can be partly described with the prescription of beam chain waviness. Instead of using a Belytschko–Schwer resultant beam formulation as in [45], which allows setting the bending stiffness of beam element arbitrarily, a Hughes–Liu beam formulation with cross section integration was applied. The bending stiffness was configured by the definition through the thickness integration rules for the beam element and independent from the tensile behavior [71,72]. In the meso-scale models, the distance between the beam element chains was identical to the distance between reinforcing yarns in real fabrics (Figure 8). The beam elements that represent the reinforcing yarns had the same diameter for both fabric variants (0.75 mm). The elements size used for the simulations was determined in a sensitivity analysis. The sensitivity analysis shows that the increment of elements does not improve the simulation results.

The models were used in the simulations of the textile physical tests. These virtual material tests help to verify and configure the mechanical behavior of the material models, which are used in the simulation. These virtual tests assure the suitability of the material models and their settings for further analysis in forming simulations. Tensile test, cantilever bending test, picture-frame-test, and friction test were simulated with macro- and meso-scale models. The simulations were performed with the software LS-DYNA (Livermore Software Technology Corporation, Livermore, CA, USA) using explicit analysis. The nonlinearity of material, geometry, boundary conditions, and contact formulation of the models limited the use of implicit analysis. The modelling methods are software independent.

## 4. Simulation Results

In the simulation of the fabric tensile test, the textile was fixed at one end while the other end was pulled until failure. Figure 9 shows the resulting stress of the macro-scale shell element model along the weft direction (y-direction) in variant 1 (Figure 9a) and variant 2 (Figure 9b). Although the macro model of variant 2 has higher maximum tensile force, it has lower stress because of its greater thickness. This is explained by the calculation of stress-strain curve, in which stress is the tensile force divided by the cross-sectional area. Figure 9c,d shows the axial forces within the beam elements of the meso-scale model shortly before complete failure. As can be seen in the pictures, the beam elements of both variants have the same axial force value but model of variant 2 has significantly higher beam chain density, so that the whole textile has higher tensile strength. As in the macro-scale model the fabric is considered a continuum, separate axial forces within the yarns are not observable as in the meso-scale model, what may be of interest for further yarn detailed studies.

Figure 10 gives the comparison of the force–strain curves between experiments and simulations for both fabric variants in the warp and weft directions. In general, the formulation of the macro-scale model allows for a direct input of the measured tensile stress–strain curves of the fabrics separately in the warp and weft directions [12]. Thus, the results of the simulation fit quite well with the experimental data. Beam elements in meso-scale models possess a linear elastic material model and the waviness of beam element chains was prescribed with the assumption that the waviness is harmonic and only along the reinforcing direction [45]. This resulted in a good accuracy of the simulated tensile force–strain curves with moderate deviation. If the non-linear tensile behavior of the reinforcing yarns (Figure 4b) can also be described correctly, the results would be improved. The force–strain curve of fabric variant 1 in both directions and fabric variant 2 in warp direction are almost identical, where the density of reinforcing yarns are the same. The fabric variant 2 has significant higher tensile strength along the weft direction, where the density of the reinforcing yarns is higher (Table 2). For the macro-scale model, the input tensile curve for variant 2 in the weft direction was calculated based on the tensile test data. The result of the macro-scale model fits well with the experimental data (Figure 10). On the contrary, the material model parameters for the beam elements representing the reinforcing yarns are the same for each model. However, the density of the modelled beam element chains of fabric variant 2 was increased as in the real fabric (Figure 8). The loop length of the beam element chains representing the knitting yarns for variant 2 was shorter as measured on the fabricated fabric samples.

The shear resistance of a fabric has a great influence on its drapability. The simulation of picture-frame tests helps to check and configure the shear behavior of the models. The formulation of the macro-scale model allows to use the shear force-shear angle curves from experiments to define the shear behavior of the shell elements during the forming process. The shear behavior of the meso-scale model instead is configured through the friction coefficient between the beam elements. Shear curves are displayed in Figure 11. The calculated shear force-shear angle curves agreed well with results from experiments for both models with moderate deviation. The wrinkle shapes in the experiment of one representative sample, which were captured with a 3D scanner, also show agreement with the simulations (Figure 12 and Figure 13). The location and the altitude of the wrinkles are generally predictable with both available models with limited deviation. 

In the cantilever test simulation, the textile stripe was modelled with the same overhang length as determined in the experiments. The textile stripe models laid horizontally with one end fixed. The gravitational acceleration was gradually applied in the vertical direction. The bending stiffness of the textile models was configured such that the free end of the textile stripe hits the 41.5° plane [66,67] when the gravitational acceleration was fully applied (Figure 14). The macro-scale shell element model was configured such that by tailoring the in-plane stiffness of each layer within the laminate (i.e., Young’s moduli and thickness of the layers) the bending stiffness of the shell elements was fitted to the reality [12]. The bending stiffness of the beam elements in the meso-scale model was fitted by configuring the integration points in the cross-section of the elements. While the macro-scale model allows one to fit the bending stiffness of textiles in every bending direction and bending side, the meso-scale model can only fit the bending stiffness in every bending direction by separating the integration formulation of beam elements in the warp and weft directions. Through the cantilever test simulation, the bending stiffness of the models is assured to be fitted with the real fabrics.

During the forming process, the interaction between fabrics and forming tools also has an influence on the final 3D preform. By applying sufficient blank holder forces, the forming defects can be suppressed [11]. The magnitude of friction induced by in-plane tensile forces is linearly proportional to the blank holder force and depends on the friction coefficient between textile and metal forming tool. Therefore, the interaction between textile models and forming tools in the simulation should also be verified. A friction-test simulation, which represents the actual test on the biaxial tensile test machine, was performed and used to validate the contact formulation. The set-up of this simulation model is shown in Figure 15. Two rigid plates were modelled with shell elements. They represent the metal pressers and have the same dimension as the real pressers, i.e., 50 mm × 150 mm. Fabrics were modelled in between the clamps on the macro- and meso-scales (Figure 15). The model input parameters are already set with the previous studies. In this simulation, only the contact method is analyzed. The contact formulation was based on the penalty approach [71,72]. The penalty contact method uses normal interface springs between all penetrating nodes of the participating contact surfaces. Therefore, the participating contact surface can penetrate each other slightly depending on the interface stiffness that couples the interpenetration with the consequential reaction forces. The interface stiffness has the same order of magnitude as the stiffness of the interface element normal to the interface. The contact force is proportional to the penetration depth of the participating nodes in contact. The computational cost for the contact formulation is proportional to the quality of participating nodes in the whole model. This method worked quite well for both macro- and meso-scale models and showed very good agreement with experimental results (Figure 16). If the interface pressures become large, unacceptable penetration may occur. There is no direct possibility to output the penetration in LS-DYNA. However, the contact force can be checked from the output database. As the contact stiffness is known, the penetration can be calculated. It was found that the penetration of elements (nodes) was smaller than the maximal deformation of the elements at the pressure 1 MPa.

The computational cost and number of elements of both macro- and meso-scale models for textile variant 1 are given in Table 4. In general, the meso-scale model has a significant higher cost than the macro-scale model. The computational costs increase exponentially due to the contact formulation, whose costs are exponential proportional with quantity of elements. Depending on the need of the application, a suitable model can be chosen. The computations with macro-scale model were carried out on a workstation with Intel i9-9900X CPU (3.50 Ghz, Intel, Santa Clara, CA, USA) with 20 logic cores and 64 GB RAM. The simulations with the meso-scale model were done on a high-performance computing cluster, which used Intel Sandy Bridge CPU with 32 physical cores (2.30 Ghz) and 8 GB RAM/core.

## 5. Model Application: Forming Simulation

The forming simulations were carried out to demonstrate the practical application of the developed models. A set of forming tools with a stamp of T-shape geometry was used (Figure 17). The textile fabric was placed between the blank holder and the female tool. The stamp formed the fabric into the female tool while the blank holder restrained the fabric. Two different blank holder forces were applied: 0 N (no blank holder at all) and 100 N. The results of forming simulation with both macro- and meso-models for the textile fabric variant 1 are shown in Figure 18. In general, the results from both models show agreement with each other with some deviations. Larger wrinkles were detected with macro- and meso-scale simulations when no blank holders were applied. Small wrinkles occurred at the edges of the geometry when a blank holder force of 100 N was applied during fabric forming. Those wrinkles were found using the macro-scale model, while the meso-scale forming results did no show a significant wrinkling. With macro-scale model, the shear angle of the whole textile can be observed, while the meso-scale model can give more details about axial force of every single yarn. By monitoring the axial force of every single yarn, the location with high risk of yarn damage can be predicted and suitable intervention can be implemented, for example adjustment of the blank holder force or using segmented blank holders.

## 6. Conclusions 

Two types of biaxial weft-knitted fabrics were manufactured and tested. Two kinds of numerical models were developed for those fabrics. A macro-scale model based on shell elements and a laminate formulation was established in order to account for the tensile, shear, and bending characteristics. A discrete approach based on beam elements was also developed. Both approaches can reproduce the mechanical behavior of the fabrics. With the help of the virtual textile physical tests, the models were validated for all important mechanical behaviors. The macro-scale model requires moderate computational cost, but most of the important forming mechanisms can be described with such an approach in a satisfactory way. When a higher description level is required, the meso-scale model can offer some more important forming mechanisms such us yarn sliding, in-plane and out-out-plane yarn buckling, yarn damage, and gap formation. However, this results in higher computational costs.

As shown in this study, both models are suitable for further research of the forming process and can be used independently as well as in combination. The forming of one example geometry was shown using macro- and meso-scale approaches for the fabric. A comparison of the numerical results to real forming experiments would provide more information on the validity of the model approaches. As computer technology becomes faster and more affordable due to achievements in information technology, yarn- or filament-based models can be more widely used in research and in the industry. Multifilament models are nearest to the reality. The use of filament-based models for large-scale simulations, however, is still limited. In the current trend, research focusses on the expansion of the utility of the classical macro-scale model and improve the discrete modelling methods. For example, an improvement of the compressibility of beam element can make the meso-scale model suitable for research where the compaction is important without adding much computational cost [73].

## Figures and Tables

**Figure 1 materials-12-03693-f001:**
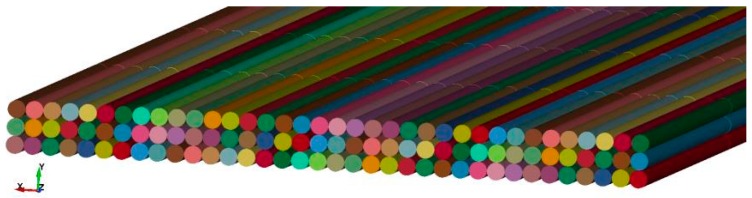
Micro-scale model of a multifilament fiber roving with beam element chains.

**Figure 2 materials-12-03693-f002:**
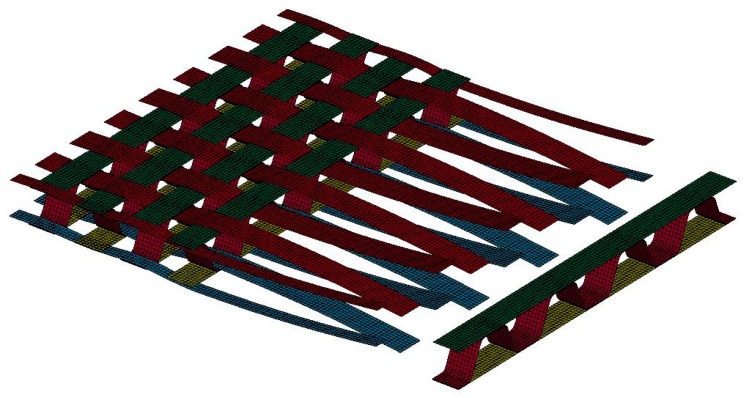
Weaving simulation of a 3D multilayers woven fabric from fiber roving, which are represented by shell elements.

**Figure 3 materials-12-03693-f003:**
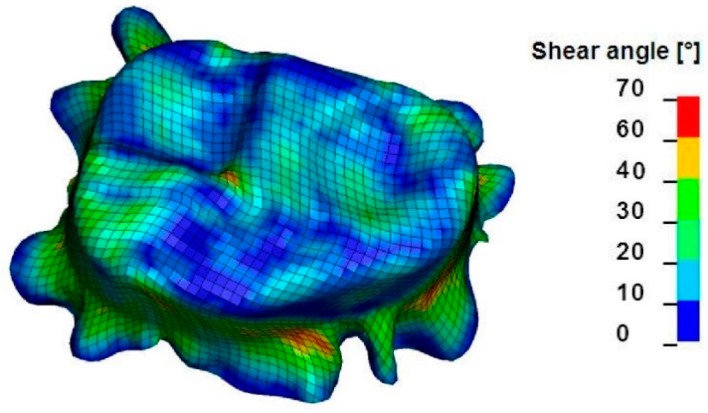
Wrinkle formation within a 3D preform simulated with a macro-scale model.

**Figure 4 materials-12-03693-f004:**
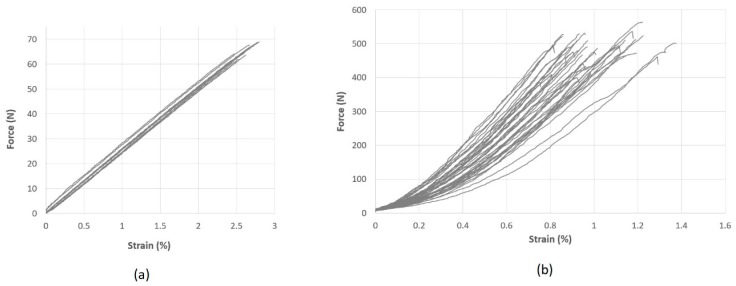
Force-strain curves of (**a**) knitting yarn glass fiber (GF)/polyamide (PA) 6.6 230 tex (10 samples) and (**b**) reinforcing yarn carbon fiber (CF)/PA 6.6 1200 tex (40 samples).

**Figure 5 materials-12-03693-f005:**
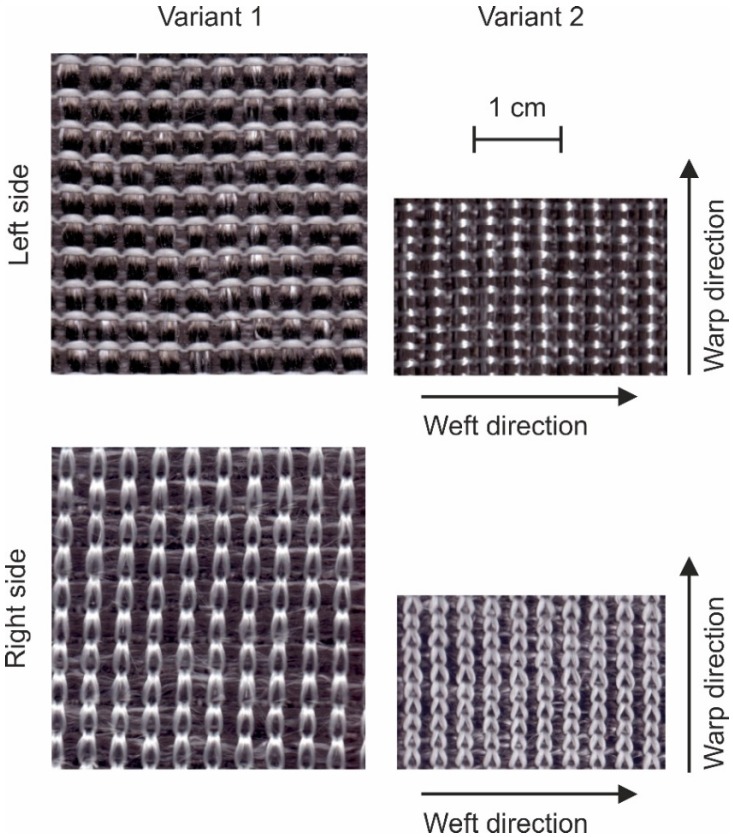
Surface of the biaxial reinforced weft-knitted fabrics with 10 warp und 10 weft yarns on the same scale.

**Figure 6 materials-12-03693-f006:**
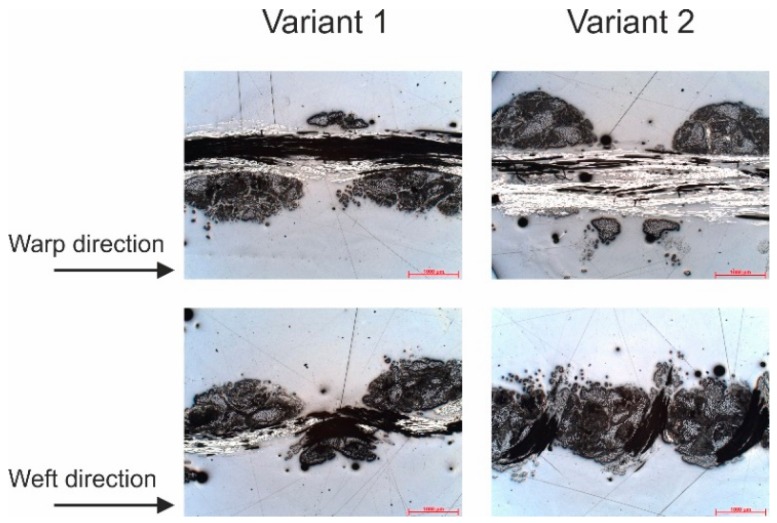
Microscopic images of the fabric’s cross-section in the warp and weft directions.

**Figure 7 materials-12-03693-f007:**
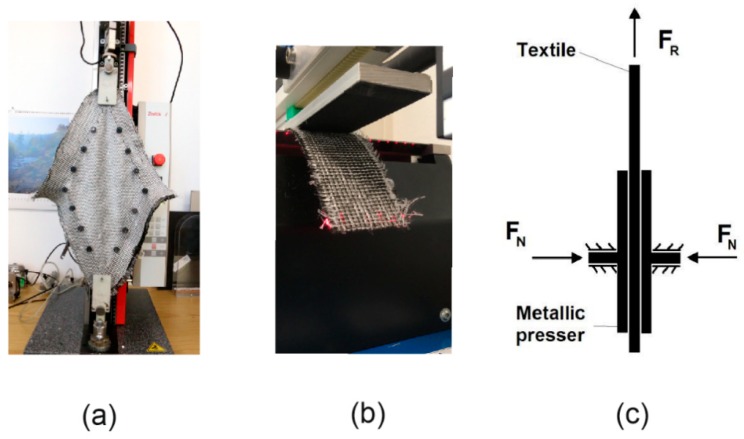
(**a**) Tensile machine with a picture frame 200 × 200mm^2^. (**b**) Cantilever test machine. (**c**) Principle of friction test.

**Figure 8 materials-12-03693-f008:**
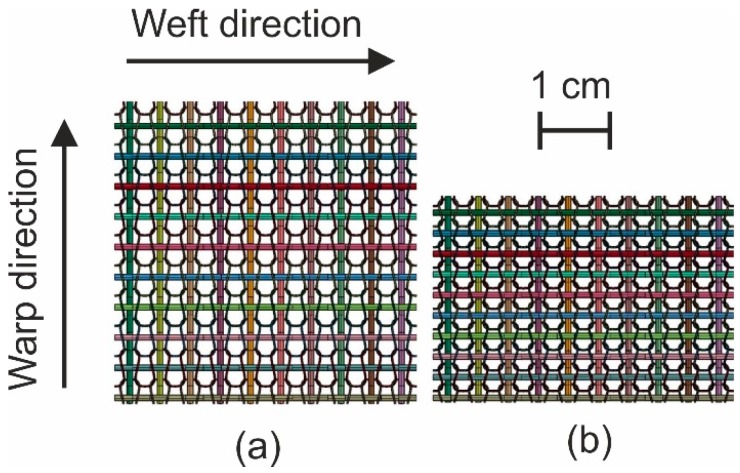
Comparison of the meso-scale models of the two fabric variants with 10 warp and 10 weft reinforcing yarns on the same scale, both variants share the same pattern but with different density (compare with Figure 5): (**a**) Variant 1; (**b**) variant 2.

**Figure 9 materials-12-03693-f009:**
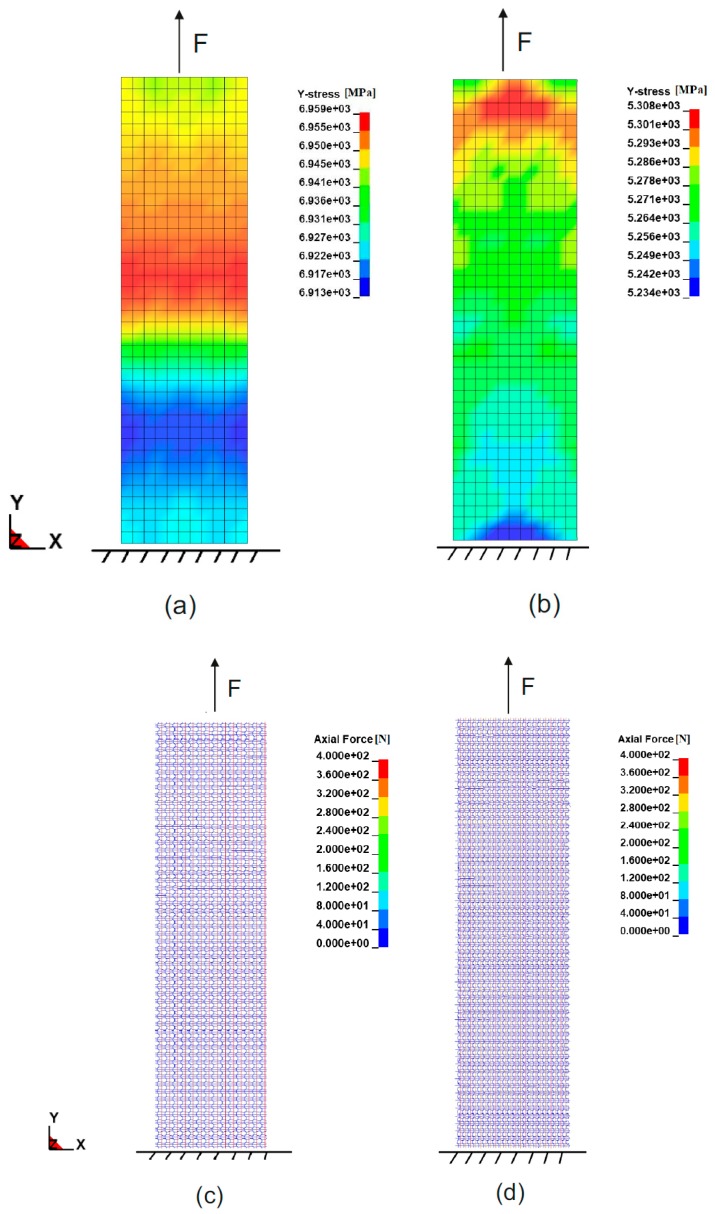
Tensile test simulation of textile fabric stripe in the weft direction with macro-scale model of (**a**) variant 1, (**b**) variant 2 (stress of textile in the weft direction presented) and with meso-scale model of (**c**) variant 1, (**d**) variant 2 (axial force of yarn presented).

**Figure 10 materials-12-03693-f010:**
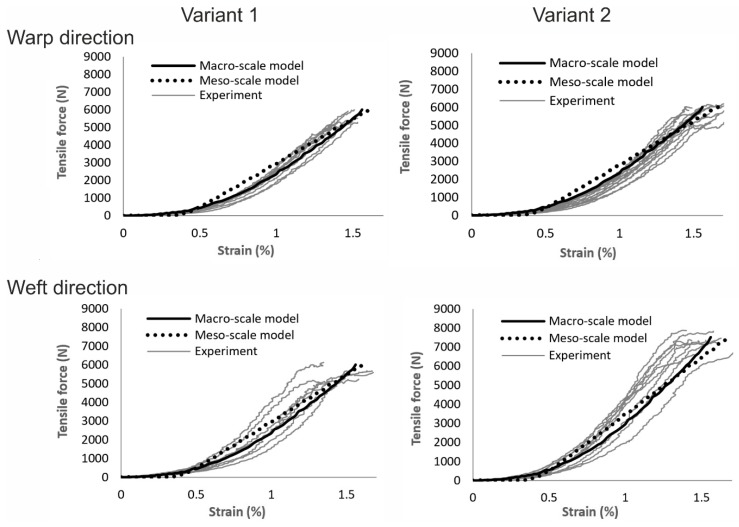
Comparison of tensile force-strain curves between experiments and simulations (16 samples of variant 1 in warp direction, 9 samples of variant 1 in weft direction, 14 samples of variant 2 in warp direction, and 10 samples of variant 2 in weft direction).

**Figure 11 materials-12-03693-f011:**
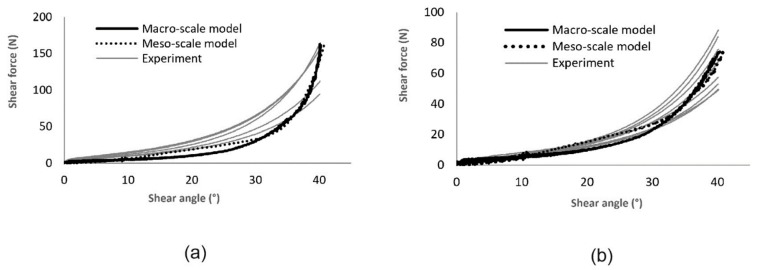
Comparison of shear force-shear angle curves between simulation and experiment of (**a**) Variant 1 (five samples) and (**b**) Variant 2 (six samples).

**Figure 12 materials-12-03693-f012:**
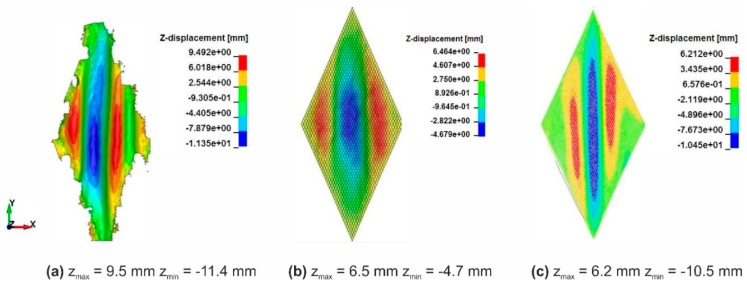
Optical comparison of the wrinkle size of fabric variant 1 at 40° shear angle: (**a**) One representative sample from experimental tests, (**b**) simulation result from macro-scale model, (**c**) simulation result from meso-scale model.

**Figure 13 materials-12-03693-f013:**
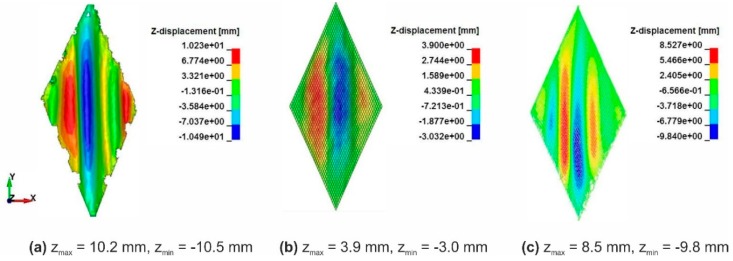
Optical comparison of the wrinkle size of fabric variant 2 at 40° shear angle: (**a**) One representative sample from experimental tests, (**b**) simulation result from macro-scale model, (**c**) simulation result from meso-scale model.

**Figure 14 materials-12-03693-f014:**
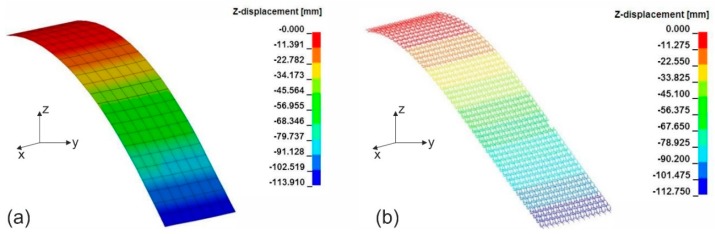
Vertical displacement of textile during cantilever test, figure showing simulation results of (**a**) macro-scale model and (**b**) meso-scale model for fabric variant 1 in the warp direction.

**Figure 15 materials-12-03693-f015:**
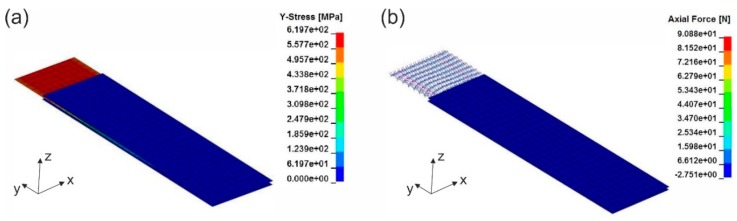
Simulation of friction test for fabric variant 1 with (**a**) macro-scale model and (**b**) meso-scale model.

**Figure 16 materials-12-03693-f016:**
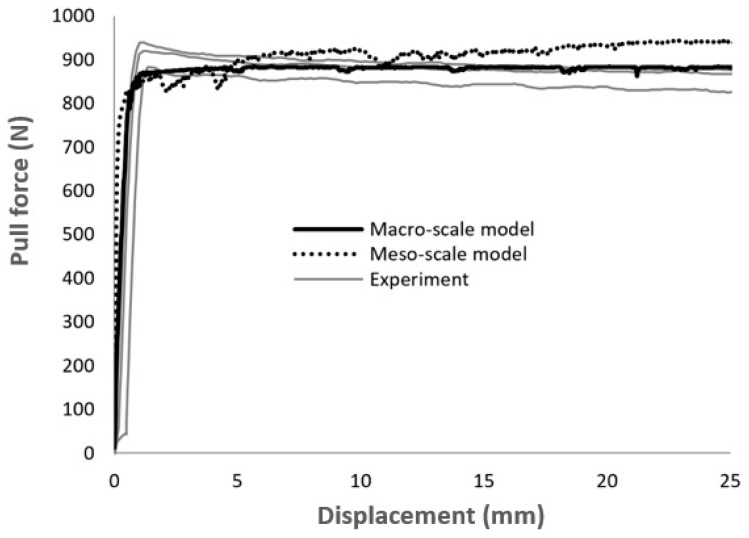
Comparison of displacement-pull force curves between experiments and simulation of friction test for fabric variant 1.

**Figure 17 materials-12-03693-f017:**
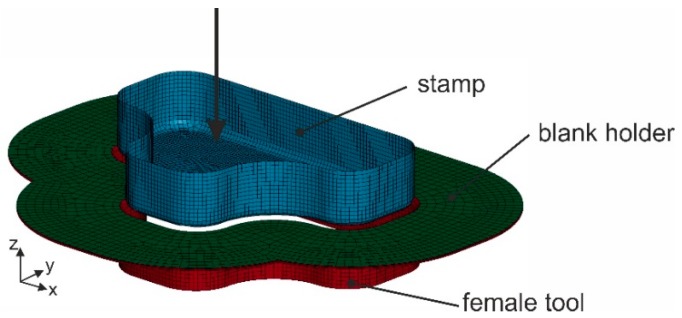
Model of the forming tool with T-shape geometry (arrow showing movement of the stamp).

**Figure 18 materials-12-03693-f018:**
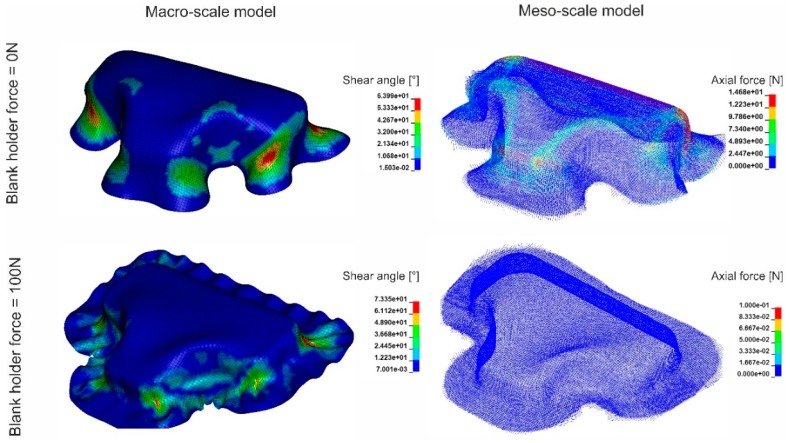
Results of forming simulation of textile fabric variant 1 with T-shape tools with macro- and meso-models at different blank holder forces.

**Table 1 materials-12-03693-t001:** Yarn properties.

Parameter	Value
Reinforcing yarn (commingled hybrid yarn and folded)	CF/PA 6.6 (4 × 300 tex)
E-Modulus of reinforcing yarn [GPa]	59.3 ± 17.7
Fracture strain of reinforcing yarn [%]	1.00 ± 0.15
Knitting yarn (folded yarn)	GF (2 x 68 tex)/PA 6.6 (94 tex)
E-Modulus of knitting yarn [GPa]	71.9 ± 6.38
Fracture strain of knitting yarn [%]	2.69 ± 0.14

**Table 2 materials-12-03693-t002:** Configuration of the two fabric variants.

Parameter	Unit	Variant 1	Variant 2
Warp yarn density	yarn/100 mm	28	28
Weft yarn density	yarn/100 mm	28	41
Knitting loop length	mm	14.4 ± 0.5	13.4 ± 0.5
Area mass density of the fabric	g/m^2^	824.2 ± 45.7	1198.8 ± 28.6
Thickness of the fabric	mm	2.22 ± 0.08	2.63 ± 0.15

**Table 3 materials-12-03693-t003:** Experimental test results of the two fabric variants.

Parameter	Unit	Direction	Variant 1	Variant 2
Max. tensile force	N	warp	5820 ± 461	5800 ± 367
		weft	5744 ± 354	7332 ± 380
Fracture strain	%	warp	1.6 ± 0.3	1.6 ± 0.1
		weft	1.6 ± 0.2	1.6 ± 0.2
Overhang length	mm	warp	171 ± 51	178 ± 14
		weft	151 ± 15	194 ± 8
Cantilever bending stiffness per unit width	Ncm^2^	warp	2.53 x 10^3^	4.14 x 10^3^
		weft	1.74 x 10^3^	5.36 x 10^3^

**Table 4 materials-12-03693-t004:** Computational cost of both macro- and meso-scale models for textile variant 1.

Simulation	Macro-Scale Model		Meso-Scale Model	
	Computational Cost (CPUh)	Number of Shell Elements	Computational Cost (CPUh)	Number of Beam Elements
Tensile stripe test	0.067	440	4	34,636
Picture-frame-test	28	1600	204	40,123
Cantilever test	16	300	1344	29,063
Friction test	0.67	300	14	25,970
Forming with T-shape tools	384	27,512	2016	237,192

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
