# Peer review of "Numerical Modelling of the Mechanical Behaviour of Biaxial Weft-Knitted Fabrics on Different Length Scales"

_materials, 2019, doi:10.3390/ma12223693_

Round 1
Reviewer 1 Report
Authors of the paper
Numerical modelling of the mechanical behaviour of biaxial weft-knitted fabrics on different length scales
present an interesting work on FE models related to knitted material made with reinforced thermoplastic yarns, including measurements of a specific case to analyze in detail.
It is quite disappointing that authors sent a review draft of poor design quality: figures repeated and wrong located, broken texts, low quality figures (5), useless figures (9), not (MDPI)-formatted reference list...
Authors make an intensive review of many of the papers about the subject related to their work, that it is itself interesting. They propose now FE models to tackle with the evaluation of reinforced knitted fabrics, considering scaled models and adequate methods. Definitely the interest for this type of materials is very high for the immediate application into industry: reinforced thermoplastics are already a must in aeronautics, and it will permeate into automobile area, etc.
Although the interest is clear, there are already many papers related to the same type of materials (glass fiber reinforced PA6). Authors do not set clear what is the net interest of this new work. To the Reviewer point of view, probably it is the application to knitted fabrics what makes the difference, because models have to adapt a non-simple geometry, the FE model has to use smart strategies to integrate complex fiber entanglements, and the final interest of calculation is a tough problem (complex geometry pieces forming). If this was the strong novelty, authors should make it clear to the Readers to stress the interest of the paper, and stress the value the paper for the Editor.
Authors made an important effort measuring samples to characterize the basics of the material and the fabrics. Tests are correctly described. However some details should set clearer, considering the experimental character and the importance of the results to assess the reliability of the models.
Authors propose two FE models, developed at meso- and macro-scales that can describe the tests and be evaluated with the measured data. If successful, the FE models would be capable to deal with the analysis of the forming processes, which is the ultimate interest. Therefore it is very important that authors state clearly the procedures used and checks done to assess the reliability of the models.
Authors should make a more detailed description of the model in some aspects, as requested, at least. Indeed, many of the missing details are already in a different paper by some of the same authors, but the Reader is not directly driven to that information.
The comparison of the models with the test results allows to verify the capability of some aspects of the models, and to tune some tricky parameters to make a more realistic model for further applications. At this point the work ends, without any application or discussion of the models compared to a real application (forming). The Reviewer has no strong problem with that and it may be well beyond the scope of the paper.
However it deprives the Readers of any glimpse of the capabilities of the model in an applied case. And this can be a really weak point. If one of the aims of the paper was to assess the capabilities of the macro- and meso-scale models to evaluate the process with fabrics, the evaluation should continue. Since Authors already have a previous work about forming using the meso-scale approach, they should consider to present a similar (or same) analysis with the macro model now, and then show the performance. This would really strengthen the content of the paper, and make a full reference paper about the subject.
If Authors consider that the new valuation (macro) and the comparison with the previous work (meso) is a difficult task to accomplish at this point, at least mention directly the results obtained in the previous paper for the Reader to evaluate the expected capabilities of the presented work.
The previous and next comments are done positively. Reviewer appreciates the effort and equality of the measurements done as well as the expertise needed to make effective FE models as those presented. The fact that the topic is related closely to industry demands, is an extra and key aspect of the paper, increasing the value of the research work.
Reviewer strongly encourages the Authors to check the draft and make an improved version that will be of interest for Materials readers.
TEXT POINTS to check:
*** Use the editorial style everywhere (esp. in References)
L33
... lower weight BY the same or enhanced ...
replace : WITH
L77
...The models could be classified...
just for completeness: refs 44 and 49 are not classified. maybe it is not possible in the 3 types scheme (?!)
FIG-4, 10 and 11
there many lines corresponding to measured data.
the caption should mention that, and mention also the number of test or samples in the figure. the Readers can have a glimpse of the spread observed in the data.
FIG-5 and FIG-8
remove repetitions, relocate and check the captions. FIG-5 the quality is rather poor. Readers can hardly see any detail of interest, and no way to compare with FIG-8 pattern. Authors show much better quality pictures of similar fabrics in other papers.
L208
...tensile testing machine Zwick Z100 with a nominal 208 load up to 100kN...
However in fig-10 the data is in the lowest 10% range of the scale.
On the one side, Authors should mention the accuracy of the measured data (force or displacement or strain). At least the full-scale-range accuracy the machine. It is not critical in the study, and the data spread is probably much bigger than any accuracy. But some reference (5% ? 10%?) is necessary to make any statement as measured data.
L209
Samples were 300 mm long and 50 mm wide and the clamping length is 200 mm...
Some dimension seems wrong (?!). 20 mm length in the clamps ?!
and the mentioned length is in one or both clamps ?
L221
The shear resistance of the fabrics was tested with a picture frame with...
The reader may not be familiar with the type of tests for fabrics. Re-write with an extra help, kind of:
The shear resistance of the fabrics was tested with a picture-frame test with a reference surface 200 x200 ...
L224
The picture frame was attached to a tensile test machine of Zwick 224 with a nominal load of 2.5kN...
and vales in fig-11 are in the lowest range.
Same comments as for L208 above.
L225 sensor
The shear force was measured with a sensor...
Same comment as for L208 about accuracy of sensors used for measuring length and force.
L232 about the 3D scanner
Same comment as for L208 about accuracy of sensors
of course the overall accuracy of a 3D scanned image is more complex, but at least the reader can appreciate the quality of the measurement done.
L226 & L227
... next to the upper CLAIM of the tensile test machine...
typo? CLAMP ?
L236
While variant 1 has great shear resistance, variant 2 requires significant lower force at the same shear angle...
is not the same information just repeated ?????
(while A is bigger than B, B is smaller than A...)
maybe Authors want to point out another information ?
overall ranges ? stiffness ?
FIG 12 & 13
add labels (a b c)
complete captions with useful information.
L253
The testing results suggested ...
why suggested ? values are very different.
only if data uncertainty or data spread was too high, the discussion would be loose. Otherwise the results are clear (!?)
L256-ff
Same comment as for L208 about accuracy of sensors
L256-ff
Authors mention static and dynamic friction coefficient measurement, but do not comment about how they proceeded. It is simple and clear looking at fig-16. but al least mention so the Reader follows the discussion.
L260
... and 4 cross contract axes
odd description (?!). The Reader can imagine the setup, but the description should not be confusing. Please re-write.
table-3
for Cantilever bending stiffness the unit is set to: cNcm2
is customary for this field?
considering the range of the measured values: way not using Ncm2 and keep homogeneous the unit used in the Table ?
section-3 (in general terms...)
authors use LS-Dyna as calculation tool, a reference tool for FE research. However at this section Authors should provide enough information as to allow Readers to reproduce the results. Also to appreciate that the models included details and checks to make them reliable.
# implicit? explicit? LS-Dyna is well renowned as THE tool for explicit evaluations. The fact that this point is not set, induces a natural question to the Reader. Just set it is (or not ?!) used as implicit.
L289
... elements, which exceed this strain limitation in any direction, were automatically deleted...
This option is tough. It is really valid if the material is or can be considered as brittle. Please state clearly that that this behavior is realistic for yarns.
L292
... tabled values...
odd expression . change.
L301-ff
The whole paragraph about the method details cannot be understand without making a direct reference to the previous Author's work
IOP Conf. Series: Materials Science and Engineering 406 (2018) 012026
It is not necessary to repeat information here, if details are described elsewhere. Therefore Authors should choose: either to repeat the necessary details in the paper to make it autonomous; or to lighten the text by referring to other paper.
L320-ff
The macro-scale models have the same mesh size of shell elements for both fabric variants despite the different reinforcing yarn density...
This comment is in the same line as above.
Readers should understand that Authors made a careful analysis of size sensitivity to ensure that keeping the same size, not odd effects appear because the density changes. Otherwise, state why it should not be of influence (maybe the size is adequate for a broad range of geometrical parameters of the knitting loops... ?!)
L320-ff
This paragraph is dedicated to the intro of the two FE models (meso and macro) as tools to analyze the performed tests. Therefore, the line L326:
...The beam elements that represent the reinforcing yarns have the same diameter for both fabric variants...
seems out of place, since it is a detail about the mesh-model for the versions.
Better fit that sentence in the previous paragraph, and clean up this paragraph to focus in the description.
L322
The meso-scale models with beam elements, whose density of the reinforcing beam element chains are exactly equal to the real fabrics...
The sentence has no sense. Re-write.
FIG-8 caption
She extra information would be really helpful to focus the attention of the reader. kind of:
Both variants share the same pattern but with different density. Compare with FIG-5. (and get a better quality FIG-5, as mentioned above...)
FIG 9
the figure show results of force and stress, each for one model (macro and meso-scale).
What for ?? The information should be useful for the discussion. Either select one model and show the two magnitudes, or one magnitude and both models. Otherwise, show all: two models, two magnitudes. The Reader will appreciate the extra information.
L351
This resulted in a moderate accuracy of the simulated tensile force-strain curves...
The 'moderate accuracy' is only visible in warp results, for Variant-1.
For other results, the spread of data samples includes both models successfully !
Even if the FE models are not perfect in all cases, it is important to stress the success and comment on the deviations. if Authors could comment about why warp and weft results can differ, or Variants, due to model characteristics, then it would be very useful.
L357-8
...the result of the macro-scale model fits well with the experimental data without changing the mesh size of the shell elements...
Since Readers have no many details of the FE model, anyone may wonder that a sensitivity mesh analysis is missing to ensure that the element sizes are adequate for both meso and macro scales without missing the point in one case. If not shown in the paper, at last mention if authors made a careful analysis to ensure that the quality of the elements (size, ratio aspect,...) are adequate to provide proper results.
In any case, authors use misleading information about the 'mesh size' (ELEMENT size ????) referred to shell elements and to beam elements is the modals analyzed. It is the same criticism as point above referred to the need of. better description of the FE model.
L359
... in contrary ...
change to
.... ON THE contrary ....
L366-ff & fig11
Authors comment, about both model results, that 'agreed very well'. That is tough to support. Again there are differences in Variants 1 and 2.
For Variant-1, modals are always below data and cross suddenly at the higher part of the range.
For Variant-2, it seems much better. But to distinguish the performance os the model, a change of scale in B would very useful. The absolute values (A range bigger than B range) is simpler to comment.
Authors should state more realistically the results obtained.
L373-ff
The wrinkle shapes in ... The location and the altitude of the wrinkles are predictable with both available models...
About the wrinkle patterns, please just look at the numbers. The differences are quite important, and questions raise.
Maybe the spread of samples soften the results (only one sample is shown); or maybe the limit values shown in the scale are due to 'spikes' and are not representative of the shape. Indeed 30 or 50% differences may be more than acceptable considering the nature of the information analysed. But in any case, support the statements with a wider picture, or soften the statements to avoid confusing the Reader.
L376
... the textile stripe are modelled ...
use either IS or stripeS
L376-ff
Authors state clearly that some material parameters are tuned to reproduce the measured results.
At this stage, the the model is just used to 'fit' parameters. If Authors consider that the model itself, with all the previous information input, is critical to have a correct behavior in this test, at least comment on that. This point better states that this tuning is only to include extra effects, but the core of the model is already providing the general behavior.
L391-ff
(about friction) It is not so clear what was tuned or check with the model. It seems that the only check done in the model was the contact method used to obtain the friction curve (fig-16).
Again Authors should state more clearly the procedure for the Reader. If the model input parameters are already set with the previous studies and now only the contact method is analyzed, then state that clear.
Additionally, the method has to be further analyzed to make it valid. At least check that the penetration of elements (nodes) is much smaller that any deformation of the elements. Otherwise the method does not work realistically (considering that extra compression effects are not included etc etc...), as authors know. That kind of detail has to be stated in the paper. Readers should see that carefully checks were applied to validate the models.
conclusions
the main criticism is about the comments on moderate and high computational cost, with NO numbers !!
since there are no details about the model used (number of nodes-elements or detailed methods) the Reader cannot guess what the discussion is about. CPU time, at least, provide the chance to evaluate the pros and cons of the meso and macro-models for the reader.
Author Response
"Please see the attachment."

Reviewer 2 Report
The title and the abstract represent the research well, the keywords are adequate.
The topic is interesting and up-to.date.
The introduction provides a good, generalized background of the topic that quickly gives the reader an appreciation of the wide range of applications for this technology.
The content of this research paper is adequate for the publication in highly technically reputed journal.
The chapter Methodology presents the sample preparation well and clear, all the necessary data are given in a table.
The examined structures are well presented by diagrams.
Testing methods and equipment are adequately described.
The discussion is extensive and well written, supported by statistical analysis.
Summary and conclusions are clear and concise.
The paper is suggested to be accepted for publication.
Minor revisions,
Figure 8 is repeated 2 times, check and correct it
Author Response
"Please see the attachment."

Round 2
Reviewer 1 Report
Authors have made an throughly effort to explain the requested details of the review. The draft have been improved with details that help to better appreciate the work and results discussed. Indeed Authors have include a new section to make a comparison of the models in a real case of application, what strengthens the content of the paper.
The Reviewer still has some questions for Authors, to clarify few points as described in the text. However, Reviewer believes that Authors can answer straightforwards and get a version ready for publication.
Besides the comments below, Reviewer strongly notes the confusing format of the PDF file with pictures popping out at any section in between paragraphs just repeated few times, making the reading quite annoying.
REWIEW
L186 & Fig-5-caption
15x15 yarns or 10x10 yarns ?? please check.
FIG-5
knitted material sides are named RIGHT and WRONG sides in some contexts.
is that what authors show as LEFT and RIGHT sides ?
L219
... The shear resistance of the fabrics was tested with a picture frame ...
for better guiding the Reader, maybe re-write as:
...The shear resistance of the fabrics was MEASURED with a picture frame TEST...
L252
for completeness, it is of interest to mention the number os samples used to obtain the values of bending stiffness.
...The testing results showed that the fabric variant 2 (XX samples) has greater bending stiffness in both directions than the first variant (XX samples), see Table 3.
L265-ff
...the textile stripe starts to slide between the clamps and the friction force decreases moderately. This behaviour is called kinetic friction.
to be consisten with the results observed, maybe re-write as
... the textile stripe starts to slide between the clamps and the friction force may decrease, defining the kinetic friction.
L270
no mention to load measurement accuracy is given for the Zwick 10kN machine of the friction test. and no number of samples tested. however an uncertainty is provided.
as experimental values, that information is really valuable. please complete the data as much as possible.
Table-3
units for bending stiffness: please keep cNcm2 if considered more appropriate. apologies for the point.
L288
...as pair values of strain and stress ...
maybe add
... as pair values of strain and stress, as measured.
L330
...explicit analysis typically explicit is used when inertia and dumping are of interest in the calculation.
if so (explicit), it is interesting that Authors mention, even briefly, why static (implicit) is not suited to the study.
L339-ff
the discussion about Figure-9 is quite confusing.
L339
...Figure 9a and 9b show the resulting stress of the macro-scale shell element model along the weft direction (y-direction).
Please, guide the reader trough the text:
Figure 9 show the resulting stress of the macro-scale shell element model along the weft direction (y-direction) in variant 1 (a) and variant 2 (b).
L340
...Although the macro model of variant 2 has higher tensile strength, it has lower stress because of its greater thickness.
two disconnected informations ?!
# in Table-3, it is quoted as max-tensile-force (instead of tensile strength): reader may be confused.
# the fact that the stress is found lower, is maybe due to thickness, but cannot be related to tensile strength. indeed both values act positively from the point of view of relative strength (safety).
# please, explain or re-write, and connect with the following lines !.
L242
...Figure 9c and 9d show the axial forces within the beam elements of the meso-scale model shortly before complete failure.
please add,
As can be seen in the pictures, the beam elements of both variants...
this line connect with the previous comment, and the values of table-3.
no way for the Reader to get a fluent discussion of the pictures.
As in the macro-scale model ...
maybe add at the end of the line something of the type:
...what may be of interest for further yarn detailed studies.
Figure-9 caption
mention the stress and force results, to make a clear distinction of the nature of the plots.
see comments above.
L380
...The location and the altitude of the wrinkles are generally predictable with both available models with deviation.
# maybe AMPLITUDE instead of ALTITUDE ?
# maybe add
LIMITED deviation to remark that even if present,the values are reasonable.
L421
...It was found that the penetration of elements (nodes) was smaller than the maximal deformation of the elements at the pressure 1 MPa.
why 1MPa ? wasthat value a max observed ?
if the numbers where obtained with a given input (1MPa) and scales equal for any pressure, better avoid misleading values, and simply point that
... than the maximal deformation of the elements at the evaluated pressure values.
Table-4
nice piece of data for Readers to see the real cost and values behind the calculations. still there are not basic information on the hardware used. is possible to include some data for completeness ? (CPU type, speed, RAM)
section 5
fig-18
Authors show the results of the model applied to a forming case, plotting
macro scale : shear angle meso model : axial force
but the Reader can hardly understand the connection of the information shown and the text description :
# what is expected in making a calculation with 0 and 100N blank forces ? # how to connect axial forces seen in mesh-model with shear angle as seen in macro-model ? # what details may be observed in each model (wrinkles...) what is point is showing that ? this section is a tour-de-force section (also a very appreciated section) and even if briefly, Authors have the opportunity to show what kind of details can the tackle with the different models. please, re-write.
L491
break paragraph after ...higher computational cost.
maybe starting with the Authors' findings:
As shown in this study, both models (meso and macros scale) are suitable for further research...
Author Response
"Please see the attachment."
